# Towards Federated Satellite Systems and Internet of Satellites: The Federation Deployment Control Protocol

**Joan A. Ruiz-de-Azua** [1,2,3,*], **Nicola Garzaniti** [4], **Alessandro Golkar** [4], **Anna Calveras** [1] **and Adriano Camps** [2,3]

1   Department of Network Engineering, Universitat Politècnica de Catalunya—UPC BarcelonaTech, 08034 Barcelona, Spain; anna.calveras@upc.edu
2   Department of Signal Theory and Communications, Universitat Politècnica de Catalunya—UPC BarcelonaTech, 08034 Barcelona, Spain; camps@tsc.upc.edu
3   Research Group in Space Science and Technologies (CTE-UPC), Institut d'Estudis Espacials de Catalunya (IEEC), 08034 Barcelona, Spain
4   Center for Entrepreneurship and Innovation, Skolkovo Institute of Science and Technology (Skoltech), 143026 Skolkovo, Russia; Nicola.Garzaniti@skoltech.ru (N.G.); A.Golkar@skoltech.ru (A.G.)
*   Correspondence: ja.ruiz-de-azua@jarao.org

**Abstract:** Presently, the Earth Observation community is demanding applications that provide low latency and high downlink capabilities. An increase in downlink contacts becomes essential to meet these new requirements. The Federated Satellite Systems concept addresses this demand by promoting satellite collaborations to share unused downlink opportunities. These collaborations are established opportunistically and temporarily, posing multiple technology challenges to be implemented in-orbit. This work contributes to the definition of the Federation Deployment Control Protocol which formalizes a mechanism to fairly establish and manage these collaborations by employing a negotiation process between the satellites. Moreover, this manuscript presents the results of a validation campaign of this protocol with three stratospheric balloons. In summary, more than 27 federations with 63.0% of throughput were established during the field campaign. Some of these federations were used to download data to the ground, and others were established to balance data storage between balloons. These federations allowed also the extension of the coverage of a ground station with a federation that relayed data through a balloon, and the achievement of a hybrid scenario with one balloon forwarding data from a ground device. The results demonstrate that the proposed protocol is functional and ready to be embedded in a CubeSat mission.

**Keywords:** Federated Satellite Systems; satellite networks; Internet of Satellites; Earth Observation; non-terrestrial-network

## 1. Introduction

Presently, new key enabling technologies have precluded the emergence of novel space-based applications that can satisfy current environmental, and socio-economic demands. The Horizon 2020 Operational Network of Individual Observation Nodes (ONION) project [1] identified the needs of the Earth Observation (EO) community to define the evolution of the EU Copernicus system in the 2020–2030 time frame. Specifically, applications to monitor the marine weather forecast, and marine fishery pressure are the most demanded ones to cover the Arctic changes, followed by the hydric stress monitoring (i.e., soil moisture) as a proxy of desertification, or crop yield. Moreover, the increase of climate disasters has accentuated the need for continuous monitoring and prediction [2]. As pointed out in [3], these applications can only be achieved with low latency and sub-metric spatial resolution observations. The consequence of having a satellite constellation equipped with the required instruments that provide these measurements would entail the increase of the data volume generated. Additionally, broadband telecommunications

services have experienced a boost in recent years with the emergence of the 5G paradigm in the aerospace domain to achieve global coverage and seamless integration with terrestrial networks [4,5]. This feature is remarkable when new applications with Wireless Sensor Networks (WSN) [6] are conceived for remote areas of the globe, such as the Arctic region, the open oceans, or other large inhabited regions [7]. To achieve this goal, the connectivity between space and ground segments must be improved.

Therefore, the increase of downlink opportunities is essential to enhance the downlink capacity in future missions. Distributed Satellite Systems (DSS) have emerged as an effective solution to meet these new demands. DSS are described as an ensemble of satellites that share a common objective. As presented in [8,9], different DSS architectures have been described in recent years. Some of these architectures proposed the use of satellite-to-satellite communications [10]. In this context, a satellite could communicate with a ground station using the downlink contact of another satellite, increasing thus the overall downlink opportunities. The Federated Satellite System (FSS) concept [11,12] aims at addressing this situation by promoting collaborations between satellites to share unused resources, such as memory storage or downlink opportunities. These collaborations, called federations, are established sporadically and opportunistically depending on the availability of the resources to be shared. The federation is established through Inter-Satellite Links (ISL), which are point-to-point communication links between satellites [13]. Due to the satellite movement, an ISL is characterized by existing only during a lapse of time, impacting the stability of the federation. To overcome this situation, the Internet of Satellites (IoSat) paradigm [14] expands this concept to a multi-hop scenario. In this context, satellites agree to be involved in a sporadic and temporal network, called Inter-Satellite Network (ISN). This kind of network is created from a decision of the intermediate satellite, and only when a federation must be established.

The creation of a federation is related to the availability of the resources that can be shared. In this context, these resources can be shared as services that a satellite provides to another satellite. However, these services are not continuously available, existing opportunities to consume them only during lapses of time. This opportunistic nature of the federations poses new communications challenges, such as intermittent connections or mechanisms to fairly share or trade the resources. Previous work [15] presented the Opportunistic Service Availability Discovery Protocol (OSADP) which provides a mechanism to know the available services among the satellites that can be requested to create a federation. Despite this knowledge, the protocol does not define a procedure to establish and maintain this federation. Other authors have tried to define this procedure in [16]. The proposed solution is based on the combination of current Internet technologies with Delay Tolerant Network (DTN) ones [17]. Specifically, a complete protocol stack with existing technologies is presented, proposing the Saratoga protocol [18] to exchange data as in a federation. This file-oriented protocol was originally designed to download data performing hop-by-hop transfers between satellites. The achieved results of this approach demonstrate the benefits of having federations to download data. However, this solution does not take into consideration a key concept of a federation: the service nature. As previously indicated, a service from a satellite is available during a lapse of time, according to the use of the resources. This temporal availability needs to be reflected in the protocol. Moreover, as the satellite resources are valuable and limited, their consumption must be properly negotiated and agreed before establishing the federation.

For that reason, this work presents a new protocol, the Federation Deployment Control Protocol (FeDeCoP), that deploys federations integrating a negotiation process to agree on the resources to be shared. To the best of authors' knowledge, this negotiation process has not been defined in a protocol for satellite federations previously. The FeDeCoP is based on a connection-oriented mechanism. Although connection-oriented protocols are typically not suitable for satellite networks (due to the disruption of the network), the deployment of a federation entails the use of a context that takes into consideration the service status. This protocol was experimented in a dedicated field campaign with three stratospheric balloons.

The goal of this campaign was to establish federations to share downlink opportunities and storage capacity. A dedicated hardware system has been implemented for this purpose: the Federated Satellite System Experiment (FSSExp) payload. This payload integrates the OSADP and FeDeCoP protocols. This manuscript presents the details and the outcome of the campaign. A discussion of the FeDeCoP performance is performed evaluating it in different configurations: when balloons are on ground, raising, and flying. The results demonstrate that using the proposed protocol the federations can be correctly established, and it is possible to increase the communications range using intermediate nodes.

As a global overview, this work mainly contributes by extending [19] with (1) a detailed description of the FeDeCoP protocol and its features, (2) an explanation of the stratospheric balloon campaign to conduct a proof of concept of a federation, (3) discussion of the generated metrics of different types of federations, and (4) evidence of the benefits of using the proposed protocol by evaluating the retrieved results of the stratospheric balloon campaign. The remainder of this work is structured as follows. First, Section 2 discusses the nature of the federations and presents the details about the FeDeCoP protocol. The concept of the experiment conducted with the stratospheric balloon campaign is presented in Section 3. The different scenarios evaluated in the campaign and the obtained results are discussed in Section 5. Finally, Section 6 concludes the work.

## 2. Federation Deployment Control Protocol

The FSS concept proposes the establishment of win-win collaborations between satellites to leverage unused satellite resources. In this context, a resource is an abstract object that encapsulates a system capacity, such as memory storage or computational capacity. These resources are consumed or released by the different actions or tasks of the satellite. Therefore, these tasks change the state of the resources over time, making them available during a lapse of time. Federations are established according to the availability of these resources, and the need to make use of them.

A satellite can offer its remaining resources as services that other satellites can consume. This satellite becomes thus the service provider, and the other satellites that want the service are known as satellite *customers*. These services can only be offered when they are available. For instance, a satellite can offer the service to download data only when it is in contact with the ground station. This situation poses the challenge of how to properly notify the available services. The Opportunistic Service Availability Discovery (OSAD) protocol [15] addresses already this situation by defining a mechanism to publish the services that are available in a satellite. With this publication, the satellites know the available services in the network. The protocol is focused only on defining this publication process. Therefore, no details are presented on how to deploy the federation once the notification is completed.

The FeDeCoP aims at addressing this technology gap by defining a set of directives to establish a federation. This protocol is based on connection-oriented communications, in which the customer triggers the request of the service (previously published). This communications scheme is necessary to keep a stable federation context or session, which is composed of the resource type and the quantity to be consumed, the status of the shared resources, among other parameters. Typically, connection-oriented protocols are composed of three phases that characterize the entire interaction. In the FeDeCoP case, these phases are known as the negotiation, the *consumption*, and the *closure* phases.

Figure 1 shows the state diagram of the FeDeCoP with the three phases and their transitions according to external events. Please note that the *not started* and the *standby* states are defined in the implementation of the protocol. They correspond respectively to the initial boot of the protocol, and the state in which the satellite is waiting for a request or a publication. This last state is thus the default mode of the protocol. The details of each phase are presented in the following sections.

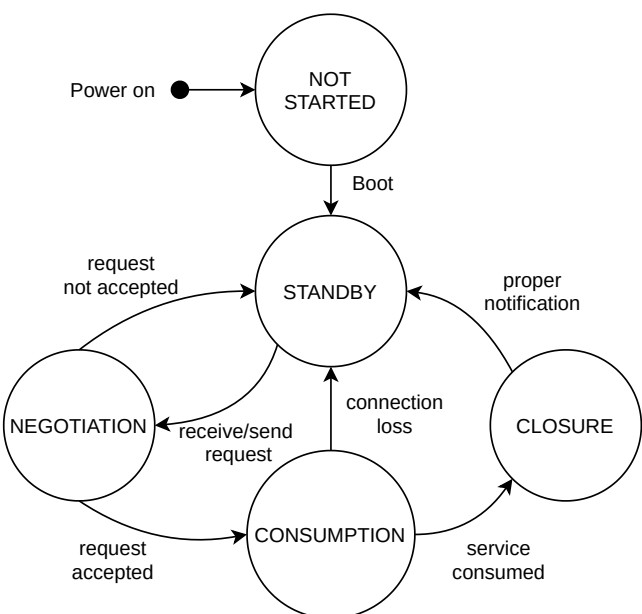

**Figure 1.** States and transitions of the FSS protocol.

## 2.1. Negotiating the Federation

The negotiation phase is triggered when a satellite requests a service. As presented in [15], the publication of the service availability is conducted using a broadband transmission. Once this notification is received by a satellite, this one sends a `request` packet indicating its interest in this service. This packet is transmitted—in a unicast manner—to the service provider, which decides to establish the federation depending on the information included in the packet. This information corresponds to the conditions that the remote satellite aims to have during the federation, and it is mapped to a set of fields. Figure 2 presents these fields integrated in the `request` packet.

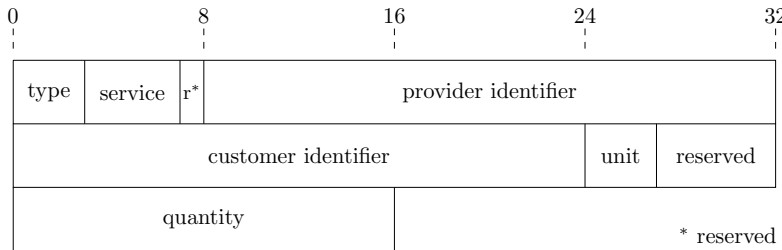

**Figure 2.** `request` and `accept` structures (unit = 1 bit).

At the very beginning of the structure, three bits identify the packet type, being in this case the `request` one. The *provider identifier* field is included to select the proper satellite if multiple providers are publishing in the network. The following *customer identifier* field determines the satellite which is requesting the service. Please note that both identifiers are represented with three bytes, which follows the format defined in [15]. After identifying the provider and the candidate customer, the type of *service* is determined with four bits. With this field, it is possible to recognize multiple services from a single provider.

Finally, the fields *quantity* and *unit* determines how much of the service is requested. In particular, the three bits of the *unit* field determines the magnitude of the value represented in the *quantity*. With this structure, it is possible to request a specific value of a resource associated with a service, such as bytes or time. For instance, a customer that would like to consume the service to store data, it would request to store 10 GB indicating the value 10 in the *quantity* and the GB unit in the *unit* field (properly coded). Please note that six bits are reserved for future extensions related to the number of services or unit levels.

The negotiation process is conducted as follows. Once a satellite receives a publication of an available service, it requests the service indicating the number of resources that would like to consume. The provider evaluates if this quantity can be satisfied. If this request can be properly satisfied, the provider replies with an `accept` packet, which follows the same structure presented in Figure 2 with the corresponding packet type. This packet includes the quantity that will be served. This value can be the requested one, indicating that the provider accepts all the conditions of the customer. Alternatively, the value can be reduced to a magnitude that is considered acceptable for the provider.

After receiving the `accept` packet, the candidate customer replies if it accepts the conditions offered by the provider. Otherwise, it discards the packet. When this second `accept` packet is received by the provider, this one considers that the negotiation is finished satisfactorily, and thus the federation is established. Please note that the packet exchange during the negotiation phase follows the well-known three-way handshake procedure.

The negotiation of the federation can be abruptly stopped if one of the `accept` packets is lost. Therefore, a retransmission process is conducted if no reply is received after a configurable timeout. After three retransmissions, the satellite considers that the link is broken. In this case, the negotiation phase is finished unsatisfactorily, and thus the satellite returns to the *standby* mode.

## 2.2. Consuming the Service

The completion of the negotiation phase leads to the proper consumption of the service, which corresponds to the consumption phase. The behavior of this phase depends on the type of service which is served. For instance, a federation for downloading data would be based on only exchanging data, while another for computational capacity would exchange a task or software code, and then wait until it is computed. The definition of the procedure to serve resources related to data and time is presented in the following paragraphs.

Federations based on exchanging data are characterized by constantly transmitting this data from the customer to the provider. This unidirectional data flow is evident in federations to download or store data. In this case, the customer transmits a `data` packet to the provider. This packet is structured as in Figure 3, with the same initial seven bytes as the `request` packet. After these initial fields, a *length* field of two bytes anticipates the number of bytes that includes the *data* field. A *checksum* is included at the end of the packet to verify the integrity of the packet itself.

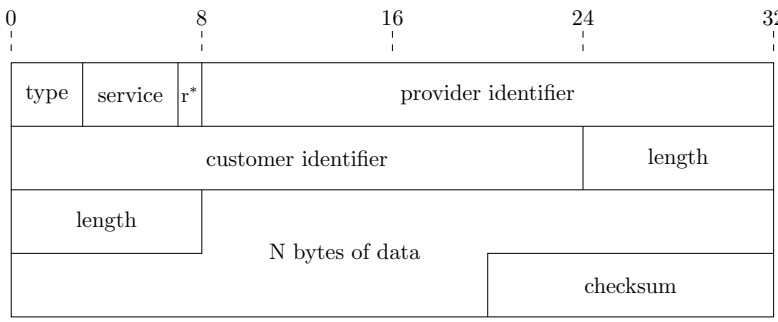

**Figure 3.** `data` structure where N corresponds to the data length (unit = 1 bit).

When the provider receives this `data` packet, it performs the pertinent action associated with the service, such as download or store. When this action is correctly achieved, the provider notifies the customer with an `acknowledgment` packet. This packet is composed of only the initial seven bytes of the structure presented in Figure 3. Please note that this transmission mechanism follows the stop-and-wait Automatic Repeat-Request (ARQ) procedure. Although this mechanism does not optimize the communications capacity of the connection (i.e., bandwidth-delay product), this process ensures that packet-by-packet the customer is aware of the completion of the service. Furthermore, the provider is al-

ways verifying that the amount of data received from the customer does not exceed the negotiated one. If it is the case, the provider would start the closure phase, detailed in Section 2.3.

The interaction between the satellites changes a little bit for those federations of time-based resources, such as computational capacity. In this case, the customer sends a `data` packet in which the *data* field corresponds to the task or the software code that needs to be executed. After the proper acknowledgment from the provider, the customer should wait the corresponding time previously negotiated. To evaluate the proper status of the service, a `keep-alive` packet is periodically transmitted by the customer. The packet structure follows the initial seven bytes of the one presented in Figure 3. With these mechanisms, the federation can remain alive during all the time that the service is being consumed in the provider (e.g., executing the code). Once this is done, the provider replies with a `data` packet including the results of the execution.

In both cases, the completion of the federated phase triggers the execution of the closure one. This phase can thus be started by the customer indicating that it has been satisfied, or by the provider indicating that the service is no longer available.

*2.3. Federation Closure*

The closure phase aims at returning the satellite to its initial state, releasing the services that were used during the federation. As this phase is triggered by one of the nodes of the federation, a three-way handshake is performed to close the federation. The exchange of `close` packets indicates the intentions to terminate the federation. This packet follows a similar structure to the previous ones (see Figure 4). The initial fields are the same as the other packets, changing the *type* value according to a `close` packet. Additionally, this packet includes an *acknowledgment* bit if it is required to acknowledge a previous message, such as a `data` packet. Therefore, the satellite that is initiating the closure phase can also notify the proper reception of the last packet with this bit. Seven reserved bits are included for future extensions.

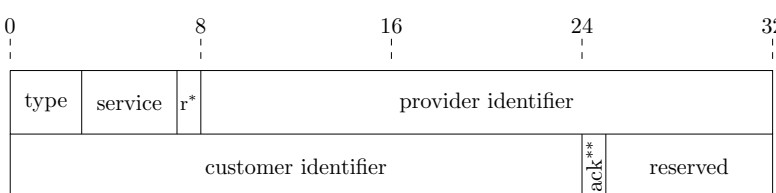

\* reserved \*\* acknowledgement

**Figure 4.** `close` packet structure (unit = 1 bit).

The packet exchange in the closure phase is performed as follows. A satellite can decide to close the federation because the service is no longer available (i.e., the provider decides) or the customer has already been satisfied (i.e., the customer decides). In both cases, the satellite that wants to terminate the federation transmits a `close` packet highlighting this intention to the other one. When the packet is received, the satellite starts releasing the resources allocated for the federation, and replies with another `close` packet which has the *acknowledgment* flag activated. With this flag, the originator of the federation closure understands that the received `close` packet is at the same time the reply of the first one. Finally, it replies with the last `acknowledgment` packet (previously presented). As remarked previously, this process follows a connection termination based on a three-way handshake. This robust mechanism to close the federation ensures that both satellites can clean the internal context associated with the federation. If the communication is lost during all the three phases, each satellite will perform the process of releasing the resources and cleaning the context after detecting that the connection is broken.

## 3. Stratospheric Balloon Campaign Overview

As part of the verification plan of the FeDeCop, a stratospheric balloon campaign was conducted. This campaign was designed to evaluate the performance of the proposed protocols and their flexibility to deploy federations with different services. In particular, the main types of federations that have been deployed were (1) to download data from a balloon to another (i.e., download service), and (2) to balance stored data between balloons (i.e., storage service). Three stratospheric balloons and a ground station were used to test these characteristics of the proposed design. From now on, the balloons are identified as Balloon A, Balloon B, and Balloon C. Each of these balloons was equipped with a model of the FSSExp payload [19,20], properly integrated for the stratospheric conditions. Figure 5 presents the payload and the subsystems that conform it: (1) the Radio Frequency Inter-Satellite Link (RF-ISL) board, (2) the On-Board Computer (OBC), and (3) the Long Range (LoRa) payload (related to another experiment). The RF-ISL board is a device that is used to establish the needed communications links between the stratospheric balloons, and the downlink contact with the ground station. It provides a half-duplex interface working at 437.35 MHz and 58 kHz of bandwidth. A Gaussian Minimum-Shift keying (GMSK) modulated signal is transmitted with 31.2 dBm of power, and it transports packets coded with Reed-Solomon. Its reception gain is 12.5 dB, and it has a 230 K of system temperature. The transmission rate is 9.6 kbps. Apart from the FSSExp, the balloons were equipped with additional subsystems, such as a battery, a Global Positioning System (GPS) receiver, among others.

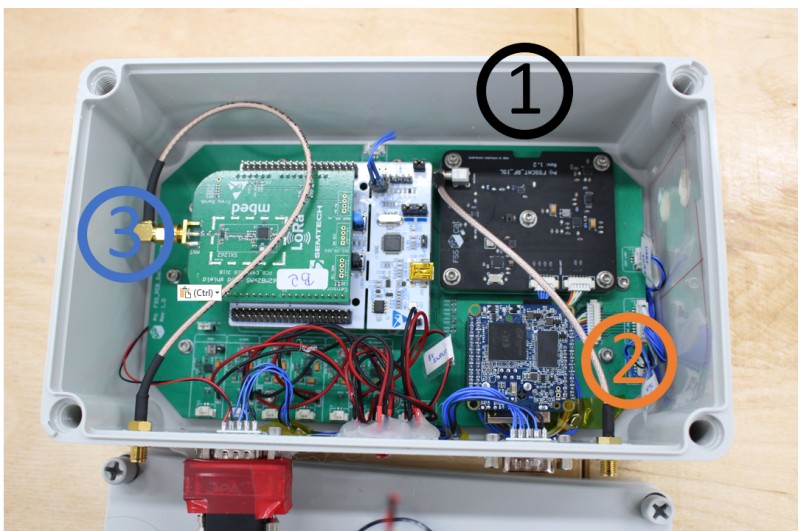

**Figure 5.** FSSExp payload with its subsystems once integrated.

The configuration between the different nodes changed depending on each federation type that was deployed. In the case of the download service, the ground station established downlink contacts with a balloon which were used to download data from other balloons. This sharing of the downlink capacity enabled the retrieval of data from other balloons that were not directly reachable from the ground station. The sequence of actions conducted in this type of federations is represented in Figure 6 and detailed as follows. First, (1) the ground station connects to one of the stratospheric balloons (i.e., Balloon A). After the connection handshake, Balloon A starts to download data until it transfers all the generated packets. At this moment, (2) Balloon A publishes the availability of the downlink contact using the OSADP protocol. This publication is received by another balloon (i.e., Balloon B), which decides to establish a federation. For this reason, (3) Balloon B transmits a request of the service, triggering the negotiation phase of the federation. Once this phase is accepted by Balloon A, the roles of both balloons are defined: being Balloon A as the service provider, and Balloon B as the customer. At this moment, (4) the balloons start the federated phase in which the customer sends its data to the service provider to be downloaded. After the

correct download, the service provider notifies the customer of the correct data transfer. This federation remains active until the contact with the ground station ends (i.e., service ending) or until the customer does not need the service anymore. At this moment, the closure phase finishes the connection, and each balloon goes back to its nominal operations (i.e., generating data). As a result, Balloon B has additional downlink opportunities to transmit data to the ground.

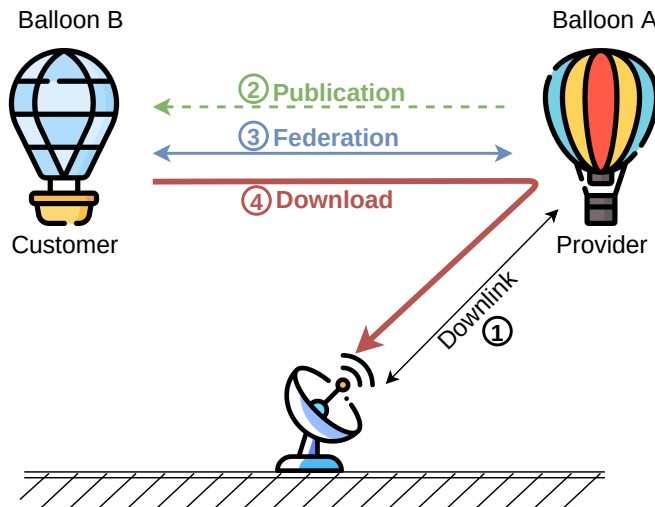

**Figure 6.** Block diagram of the subsystems and their interconnections that compose the FSSExp payload.

On the other type of federation, the balloons were able to balance the storage level by exchanging data over a federation. Unlike the previous case, no ground station was triggering the publication of the service. Instead, the balloon itself decided to publish the storage service if it detected that the storage level is below a certain threshold. Thanks to this publication, the federation was established between two balloons, and the corresponding data exchanged. Please note that in this case the federation was closed when the threshold of the storage level was reached. Using the same protocol design is possible to deploy federations with different associated services. These configurations achieved with the balloon campaign highlighted the versatility of the proposed protocol. Future campaigns may focus on the demonstration of the adaptability of the protocol to other kinds of services.

The campaign was performed in the Yaroslavl Oblast region in Russia. Specifically, the coordinates in which the launchpad was located were 56.62° N, 38.68° E. The launch started at 12:12 (24-h format) on 14 December 2019. The balloons were launched sequentially, keeping 5 min between each balloon. The wind speed on the ground was 18 km/h to the North, generating an initial distance between the balloons of 1.5 km. This distance helped to spread the balloons and to achieve a configuration that promotes the interconnection between the balloons. This connected scenario allowed evaluation of the benefits of deploying federations to communicate with those balloons that are no longer in line-of-sight with the ground station. The launch sequence started with Balloon A, then Balloon B, and Balloon C was the last one. Moreover, as the three balloons present some structural differences, the initial distance between the balloons was not kept during the entire flight (see Figure 7). This also enabled evaluation of the proposed solution in different ranges. Each balloon spent roughly 1 h and 33 min flying, landing 80 km away from the launchpad. The balloons also achieved its maximum altitude of 25 km after 1 h of the launch. This profile enabled having 30 min of contacts between the balloons and the ground station to perform multiple federations. Section 5 presents in detail these federations and discusses the results achieved.

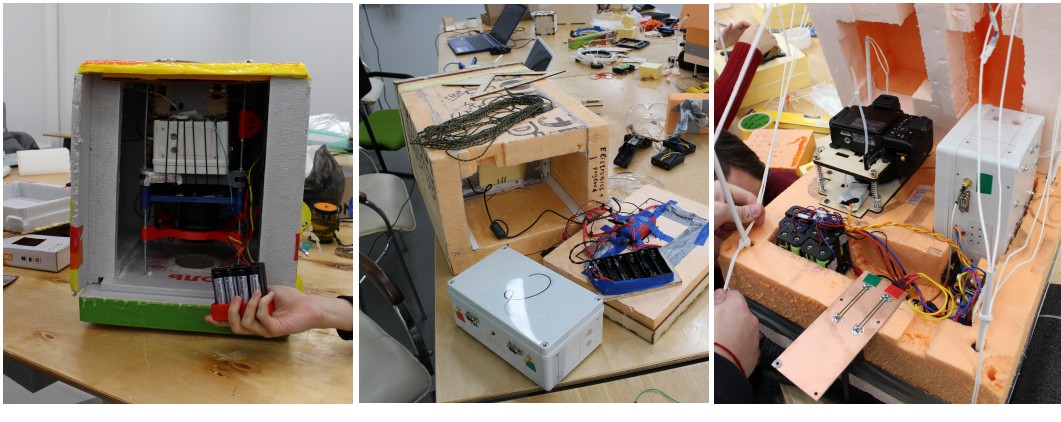

(**a**) Integrated in Balloon A     (**b**) With Balloon B     (**c**) Integrated in Balloon C

**Figure 7.** Integration of the FSSExp payload (gray box) in the three balloon structures.

### 4. Materials and Methods

The FSSExp payload software is implemented in Java language to leverage on its portability and the object-oriented approach. All the payload code is available in the project repository [21]. Additionally, this repository includes a copy of the raw data retrieved during the campaign, and the corresponding code to process it. The raw data is processed to retrieve a set of metrics that enable the evaluation of the performance of the protocols. This section presents the used metrics to measure and evaluate the performance of FeDeCoP and OSADP protocols. The former result of each federation is the sequence of packets that are exchanged. This sequence enables the understanding of whether all the phases of the federation, as well as agreed closures or unexpected ones. The loss of packets is also highlighted, which provides a view of the channel quality. This sequence is generated from the transmitted and received packets en each node, which are stored during the execution.

Another metric is the buffer usage of each balloon. As the federation types during the campaign are related to exchanging data, the need to establish a federation and the corresponding transfer is reflected on the buffer level. As in the previous metric, this level is periodically stored in each balloon, and the buffer unit is a telemetry block. This block includes different status fields, such as temperatures, power transmission, etc.

Finally, the throughput of the federation is also analyzed. This metric corresponds to the amount of data (in bits) that are received and acknowledge during a period of time. This throughput is measured per each federation, and it is compared with the theoretical maximum throughput. The definition of this theoretical maximum is driven by the average Round Trip Time (RTT). This metric corresponds to the lapse of time that a data packet needs to be downloaded and acknowledged to transmit a new one. The quality of the channel can provoke the loss of packets, and thus their retransmission. This retransmission increases the RTT because a Retransmission Timeout (RTO) is spent for each lost packet. Therefore, the worse channel quality, larger the RTT is. In this campaign, a maximum of three retransmissions have been configured. The quality of the channel can be measured with the Packet Error Rate (PER) metric, which corresponds to the ratio between the number of lost packets and the transmitted ones. This PER is related to the SNR, which is impacted by the existence of the interference signal previously identified. In this scenario, the average of the RTT is defined as follows:

$$RTT = \sum_{k=0}^{3} RTT(p = k) \cdot Pr(p = k), \tag{1}$$

where $k$ corresponds to the index of loss packets, $RTT(p = k)$ represents the RTT with $k$ packets loss, and $Pr(p = k)$ is the probability of losing $k$ packets.

Taking into consideration the ARQ mechanism, the average of the minimum RTT of the downlink (link 1 in Figure 6) can be defined as follows:

$$RTT = 2t_{tx}\left(1 - \alpha^4\right) + (t_{tx} + RTO)\left(\frac{3\alpha^5 - 2\alpha^4 + \alpha}{1 - \alpha}\right), \tag{2}$$

where $t_{tx}$ the packet transmission time, $RTO$ the timeout value, and

$$\alpha = \rho_d(2 - \rho_d), \tag{3}$$

where $\rho_d$ corresponds to the PER of the downlink channel. Please note that this definition does not include the processing time of each packet, because the minimum RTT does not consider this situation. In this campaign, the RTO was set to 6 s, and the FSSExp payload always transmits a packet of 255 bytes, by filling those that are smaller. Therefore, the transmission time is the same for all the packet types (either data or acknowledgment packets). After different measurements, the transmission time of a packet is 425 ms.

The federation interaction is composed by the balloon-to-balloon link with the downlink to the ground station (links 3 and 1 in Figure 6). The minimum RTT in this case is defined as follows:

$$RTT = 4t_{tx}\left(1 - \beta^4\right) + (t_{tx} + RTO)\left(\frac{3\beta^5 - 2\beta^4 + \beta}{1 - \beta}\right), \tag{4}$$

where

$$\beta = 1 - (1 - \rho_d)^2(1 - \rho_f)^2, \tag{5}$$

and $\rho_f$ corresponds to the PER of the federation channel.

The throughput ($S$) is the number of bits ($l$) downloaded in an RTT. Therefore, the maximum throughput is defined with the minimum RTT:

$$S_{max} = \frac{l}{RTT}. \tag{6}$$

The comparison of the measured throughput with respect to the theoretical maximum one helps to understand the limitations of the implementation, and identify possible future enhancements. In particular, with a measured PER the achieved throughput is less than the corresponding maximum one, it means that current implementation is including an additional delay that is driven the performance. This information is essential to understand if the performance is determined by the protocol design or the corresponding implementation. Table 1 summarizes the three metrics that conforms the basis of the results discussion and allows the understanding and quantification of the performance of the proposed protocols. The discussion is performed in the following section.

**Table 1.** Summary of the metrics used to perform the discussion.

| Metric | Utility |
| --- | --- |
| Packet sequence | Evaluate all the phases of a federation, and packet retransmissions. |
| Data buffer usage | Verify the conditions of a federation, and the transfer of data. |
| Measured throughput | Evaluate design limitations, and possible enhancements. |

## 5. Discussion of the Results

The execution of the FeDeCoP and OSADP protocols during the stratospheric balloon campaign enabled the establishment of different download and storage federations between the balloons. This section presents and discusses the results retrieved during the entire launch campaign. A study of the radio frequency environment in the launchpad is presented in Section 5.1. Then, Section 5.2 discusses the performance of the different federations established when the balloons are on the ground. This scenario is used as a reference for the following scenarios, in which the balloons are flying. Sections 5.3 and 5.4 present the performance of the download and storage federations, respectively.

### 5.1. Communications Conditions

Since the moment that the balloons are released, they start to move away from the ground station, reaching a maximum value of 95 km. The FSSExp payload is designed to establish a communication link between two modules separated up to 400 km. Therefore, the FSSExp payload can communicate with the ground station during the entire flight. However, other aspects may also impact this effective communication range. This communication is feasible only if the Signal-to-Noise Ratio (SNR) of the received signal is greater than the minimum required. In the laboratory, the noise floor measured with the Received Signal Strength Indicator (RSSI) from the FSSExp payload, and connecting a matched load as an antenna was $-113.0$ dBm (average). Considering the rural area of the launchpad, the expected noise floor should be $-104.7$ dBm according to the International Telecommunication Union Radiocommunication Sector (ITU-R) recommendations [22]. However, the noise floor in the launchpad was $-85.0$ dBm (average), 19.7 dB greater than the expected one. Figure 8 presents the difference between these three noise floors. The blue line corresponds to the measured noise floor, the red line corresponds to the measured noise floor in the laboratory, and the green line corresponds to the estimated one in the launchpad. This increase generates an antenna temperature of $2.9 \times 10^5$ K, two orders of magnitude larger than the expected one (i.e., 2900 K). The existence of an interference signal is the cause of this increase. The link budget is thus impacted by reducing the maximum communications range to 55 km, which entails the loss of communication between 13:00 and 13:15.

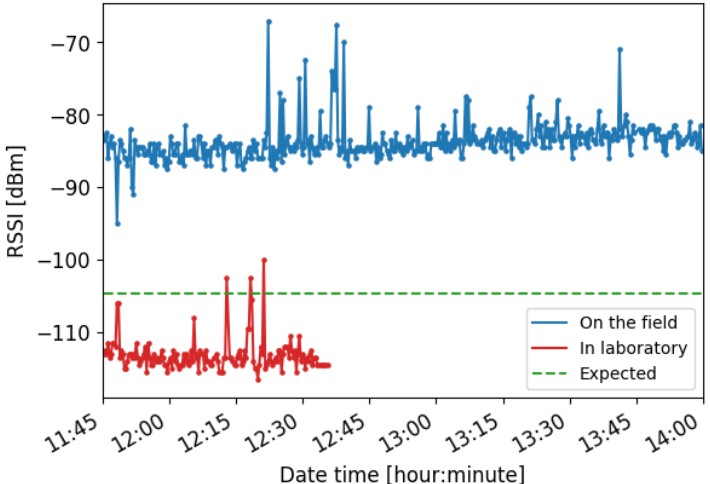

**Figure 8.** RSSI measurement of the noise floor on the launchpad (blue line), the expected one (green line), and in the laboratory (red line).

Although the increase of this noise floor, the communication with the balloon and the ground station is feasible during almost all the ascending phase. To take some measurements of the received signals, an additional Software Defined Radio (SDR) module was installed near the ground station. Figure 9 presents a short fragment of the spectrum history retrieved with this device while the balloons were flying. At the top of the figure, the spectrum of the current received signal is plotted. The presented curve shows the shape of a Gaussian Minimum-Shift Keying (GMSK) modulated signal centered at 437.35 MHz. Additionally, at the bottom of the figure, a history of the spectrum is presented over time. A color scale is used in this history to highlight the power of the signal, being orange the great values, and blue the lowest ones.

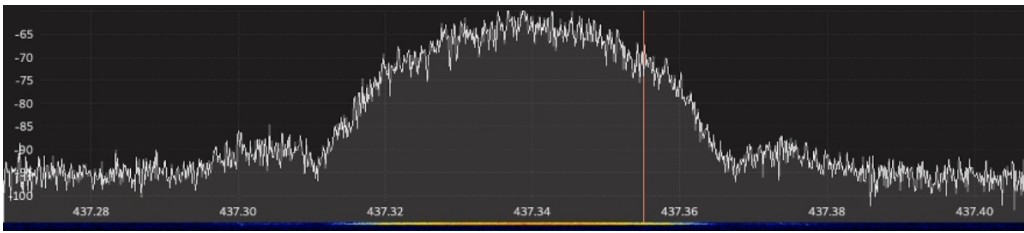

**Figure 9.** Signal sniffed from the ground station transmitted by two different stratospheric balloons during a federation.

The record of all the sniffed packets in the ground station enables the evaluation of the impact of this separation in the performance of the communications. Figure 10 shows the evolution of the SNR of Balloon A packets, which decrease over time. Before the release of the balloon, the SNR remains around 40 dB. After it takes off, the SNR rapidly goes down until 16 dB (around 12:50). Then, the communications between the balloons and the ground station is lost. Please note that this situation is coherent with the degradation of the link budget due to an interference signal, which determines that the communications should be lost after 13:00.

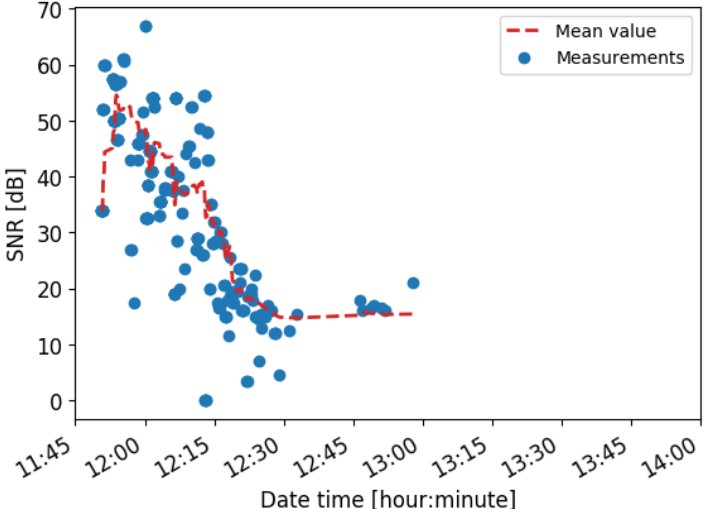

**Figure 10.** Signal sniffed from the ground station transmitted by two different stratospheric balloons during a federation.

Although the communication with the ground station is lost, the balloons keep interacting between them as Figure 11 highlights. This figure presents the SNR of the `publish` (blue dots), and the `accept` (orange dots) packets received in the balloons. Three different time slots are identified depending on their location: The former slot (purple box) is delimited between 11:53 and 12:12 when the balloons were placed on the ground. Because of the flying dynamics are not present, this scenario becomes the performance reference. The following slot (red box) is between 12:12 and 12:30, when the balloons are released sequentially. Finally, the slot between 12:30 and 12:45 (green box) shows federations established autonomously between the balloons to share storage capacity. After this moment, no longer federations were established. This situation happened because the balloons do not require more storage capacity, and they just started to publish the availability of the service. The following section presents the details of the federations established during each period.

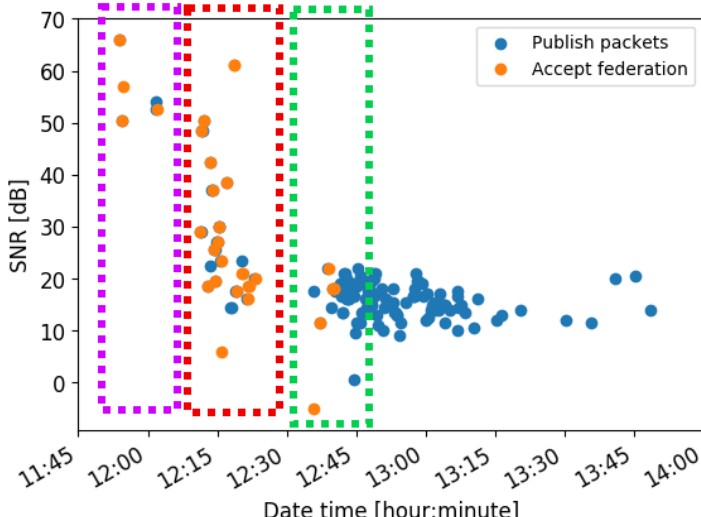

**Figure 11.** SNR of packets that establish a federation, such as publish (blue dots) and accept (orange dots) packets. The three regions are remarked with dashed boxes.

*5.2. Federations on Ground*

When the balloons are still on the ground, multiple federations were established to verify the correct operation of the payloads. Among all of them, Figure 12 presents the packet sequence of a federation. This sequence represents the different packets exchanged between the balloons and the ground station scheduled on time. Moreover, this representation identifies the types of packets using different colors and shape markers. For instance, all the blue markers correspond to packets exchanged in a downlink connection, while the red markers identify those exchanged in a federation. The rhombus (◇) and the times (×) markers represent respectively the packets to establish and close a connection (i.e., a downlink or a federation). In the case of a federation, these are the `request`, the `accept`, and the `close` packets. The dot (○) and the triangle (△) markers correspond respectively to data and acknowledgment packets. Moreover, the packet that corresponds to a retransmission is represented with an "R" below the marker. Finally, the green markers correspond to publications of the service. This kind of representation helps to understand the periods when a downlink or a federation is established with a dashed line of the corresponding color.

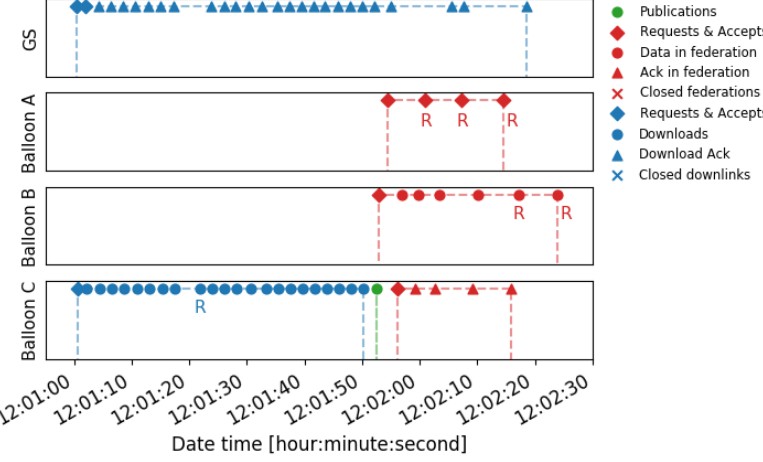

**Figure 12.** Packet sequence of federations established on ground.

Returning to Figure 12, a federation between Balloon C and Balloon B is established from 12:01:00 to 12:02:30. At the very beginning, the ground station opens a downlink connection with Balloon C, which starts to download its data for 50 s. At this moment, the

balloon has downloaded all its data, and starts to publish the availability of the download service. Balloon B and Balloon A receives this publication and decide to request the service to download its data. Balloon B reacts faster than Balloon A, being able to establish the federation with Balloon C. Although Balloon A does not receive any `accept` packet, it retransmits the `request` packet. After three retransmissions, it stops. Meanwhile, Balloon B transmits data to Balloon C which downloads it to the ground station. As part of the test, Balloon C is turned off during the federation (at 12:02:16). After that, Balloon B performs different retransmissions of the last data packet because it has not received its acknowledgment. At 12:02:23, it decides that the connection is lost, and closes the federation internally.

Up to four federation packets are downloaded from Balloon B during the federation. Figure 13 presents the memory usage of Balloon B and Balloon C. Please note that the buffer usage is measured every 20 s in each balloon. When Balloon C establishes the downlink connection, it reduces the usage to zero by directly downloading them to the ground station. When the federation is established, Balloon B reduces its usage by using the federation. Once the federation is closed, Balloon B keeps producing new data packets and thus increasing the usage of the buffer.

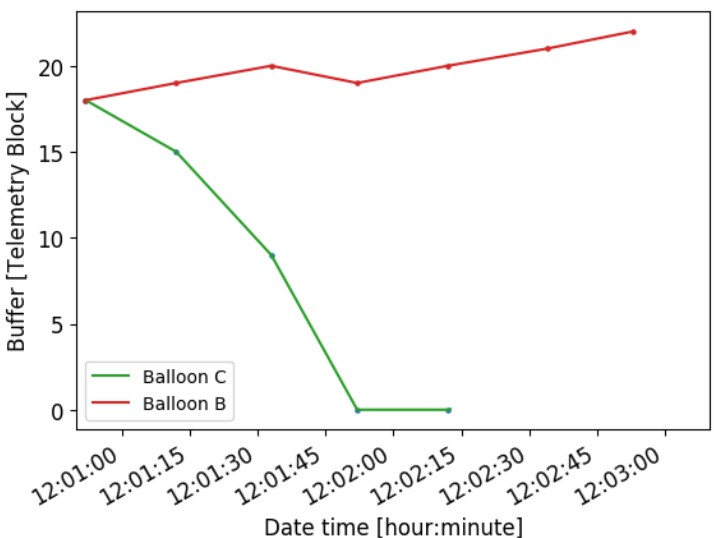

**Figure 13.** Buffer usage of Balloon C (green) and Balloon B (red) in data packet unit.

The throughput achieved with the federation is 184.7 bps, while the direct downlink has 756.9 bps of throughput. Both results can be compared with the maximum throughput according to the definition of Equation (6). In this scenario, the downlink communication has 4.3% of PER, and the federation one has 20.0%. The throughput achieved corresponds to 756.9 bps, and 184.7 bps respectively, which is equivalent to 71.0% and 98.2% of the maximum throughput with this PER. This performance indicates that the software implementation of the FSSExp payload is impacting the achieved throughput, being not possible to retrieve the maximum value in the federation case. Specifically, the processing time related to each packet increases the RTT value, which reduces the corresponding throughput. As in the federation case there is an intermediate node, this decrease is more explicit.

Apart of evaluating the impact of the channel quality, these results demonstrate that the federations can be established with the OSADP, and FeDeCoP protocols. The following section extends these results by presenting their performance with the balloons flying in different configurations.

### 5.3. Federations While Releasing the Balloons

Other federations are established while the balloons start to lift. Figure 14 presents the packet sequence of a set of them when Balloon A is raising. In this case, the customer of the

federation is Balloon B which requests up to eight federations in more than three minutes. Specifically, the ground station opens the downlink connection with Balloon A at 12:12:46, which starts to download its data. Then, it publishes the service to the others. Balloon B requests the service at 12:12:54, which is suddenly accepted by Balloon A, starting the first federation. Balloon B downloads its data using the federation, which remains open until Balloon B empties its buffer (at 12:13:20). At this moment, the federation is no longer needed, and Balloon B triggers the closure phase. After the correct three packet exchange, the federation is closed, and Balloon A starts to publish the service again (because the downlink is still open). Please note that Balloon A keeps downloading its data while it is publishing the service.

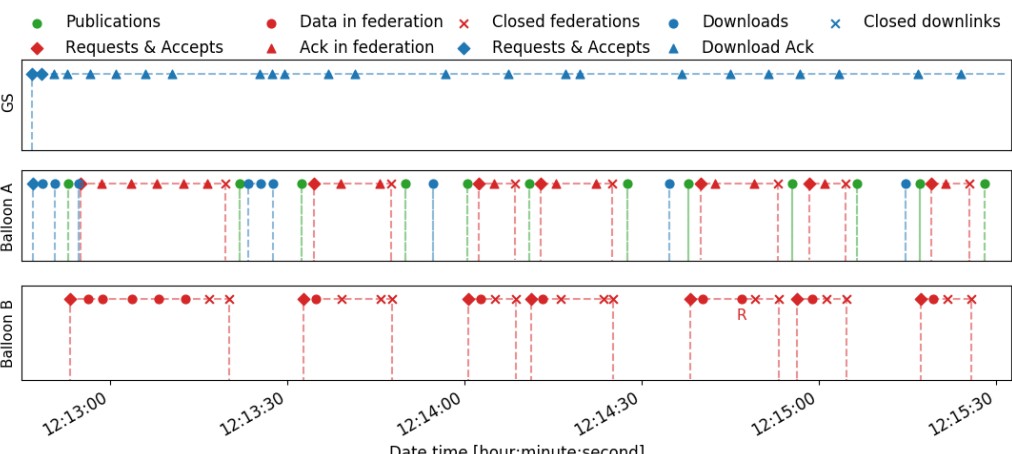

**Figure 14.** Sequence of packets exchanged in federations while the balloons start to be released.

A new service request from Balloon B is performed at 12:13:32, because it has generated an additional data packet. After downloading the packet, Balloon B closes again the federation. This situation is periodically repeated due to the data generation of Balloon B. Please note that during all these federations, Balloon B is placed on ground, while Balloon A is flying. This means that Balloon B can download data to the ground station using the relay of Balloon A which is indeed over both nodes, as Figure 15 shows. From the first federation until the last one, Balloon A follows a trajectory that reaches elevation angles close to 90°. This high elevation provokes the retransmission of some packets that are lost. It is the case of the last federation, which indeed the connection is not established. Therefore, asymmetric links exist in this scenario.

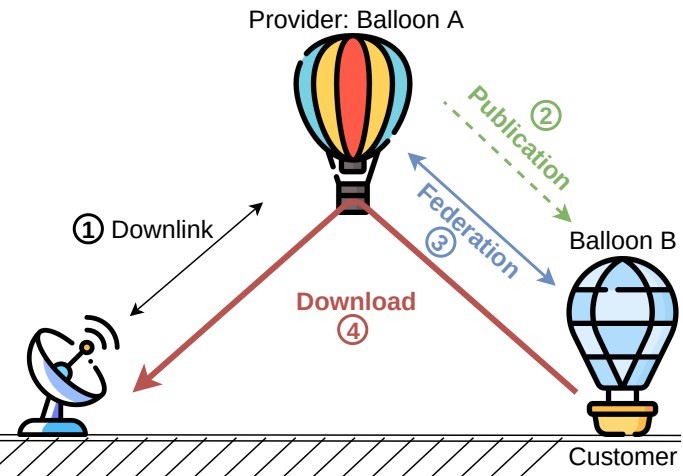

**Figure 15.** Representation of the federation established while balloons being released.

The buffer usage, presented in Figure 16, highlights this data download with the federation. When the downlink is established, Balloon A downloads all its data and empties its buffer. Then, Balloon B can also empty its buffer thanks to the federation with Balloon A. After that, both buffers remain empty thanks to the multiple federations. Once the communication is not feasible, Balloon B increases its buffer with new data (at 12:15:30).

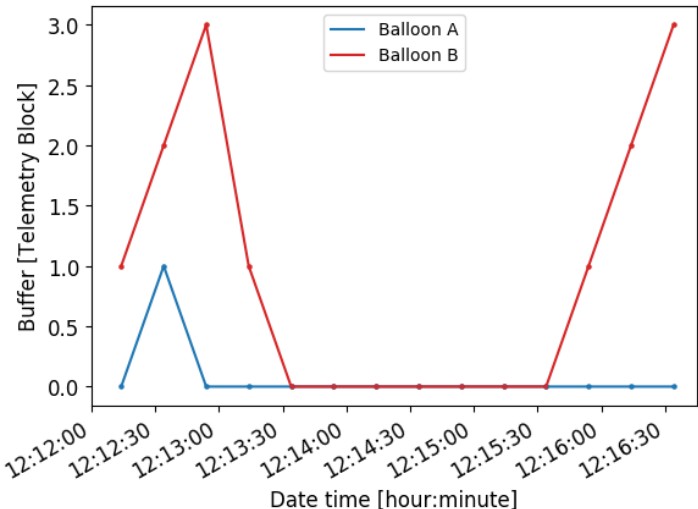

**Figure 16.** Buffer usage of Balloon A (blue) and Balloon B (red) in data packet unit while balloons are released.

The global throughputs achieved in this scenario remains similar to the ground case, being 790.6 bps for downlink communications and 167.4 bps for federation communications. This performance is achieved because the PER also remains similar, with 0.0% for downlink, and 15.6% for federation. This throughput is equivalent to 43.8% and 63.0% (respectively) of the maximum throughput, defined in Equation (6).

After finishing this interaction, Balloon B starts to fly. At this moment, the communications between Balloon A and the ground station experienced a degradation in the link quality, making it difficult to keep the connection open. For that reason, the operator switched to Balloon B to download data from Balloon A, as represented in Figure 6. Figure 17 presents the packet sequence during this situation, demonstrating the establishment of multiple federations that enabled downloading Balloon A data. As in the previous cases, the ground station opens the downlink connection with Balloon B at 12:18:40, and this one starts to download data. After finishing, it starts to publish the service, which Balloon A requests periodically. These federations help to connect Balloon A with the ground station (over Balloon B), which the communication was no longer feasible. Please note that some packets are retransmitted due to the asymmetric link created between balloons, as in the previous case.

Although those communications difficulties, the federations enable downloading data from Balloon A, being able to properly empty its buffer. Figure 18 presents the buffer usage of both balloons during this scenario. Note how Balloon A empties its buffer using the different federations, until the communication is no longer feasible with Balloon B. This communication loss is mainly caused by the high elevation angle between them.

In this case, the achieved throughput changes between the federation and downlink communications to the previous scenarios. In particular, the downlink remains during all the time with a PER of 0.0%, which enables achievement of a throughput of 621.2 bps, i.e., 34.4% of the maximum throughput. Meanwhile, the PER of federation communications is 38.5%, larger than in the other cases. Therefore, its throughput also is reduced to 96.2 bps, which corresponds to 49.2% of the maximum throughput (Equation (6)).

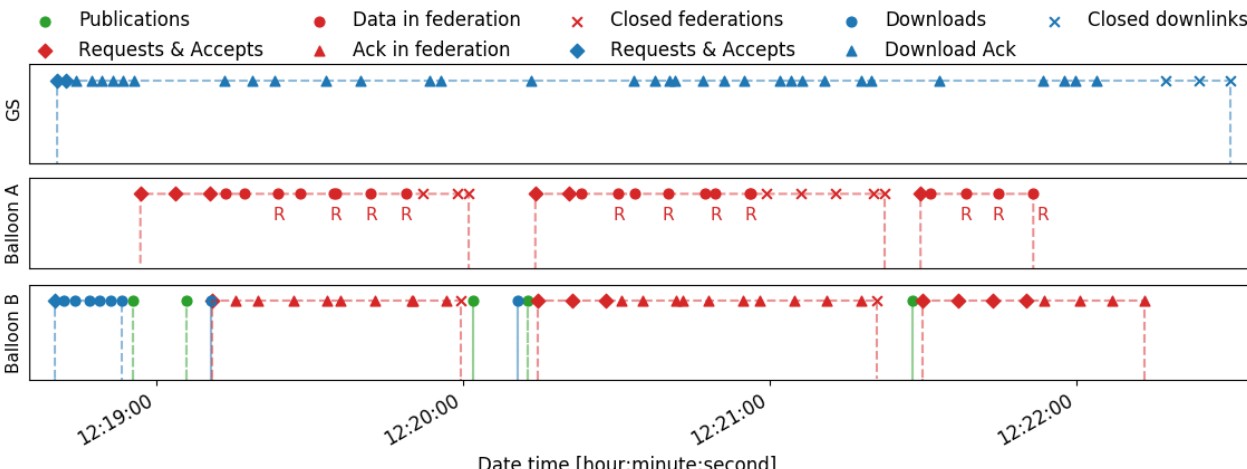

**Figure 17.** Sequence of packets exchanged in federations with a balloon as a data relay.

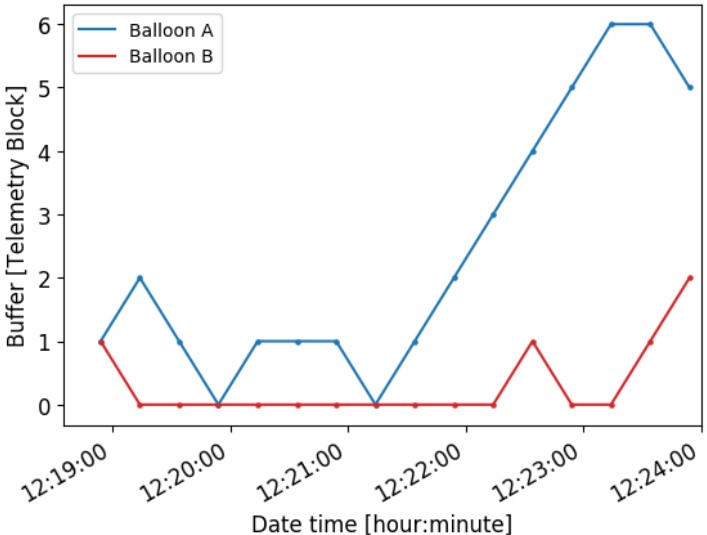

**Figure 18.** Buffer usage of Balloon A (blue) and Balloon B (red) in data packet unit with a balloon as a relay.

### 5.4. Federations to Share Storage Capacity

After accomplishing the federations to share downlink opportunities, federations to share storage capacity were the target. At 12:35:26, Balloon A started to publish the availability of this service (Figure 19). This publication is received by Balloon B which requests the service to store 38 data packets. The request is accepted by Balloon A, which provokes the establishment of the federation. This situation is repeated up to four successful federations. The first federation remains open for 1 min and 13 s, the following one for 1 min and 25 s, the next one for 52 s, and the final one for 27 s. This decrease is caused by the degradation of the link quality. Specifically, the asymmetric link between Balloon A and B (as in previous cases).

Although these difficulties, multiple data packets are exchanged between the balloons. Figure 20 presents the buffer fluctuation in each balloon during this scenario. Balloon B starts to transfer data to Balloon A, which enable a decrease to the buffer usage. Meanwhile, the buffer usage in Balloon A increases, because it is receiving the data from Balloon B. Please note that there are lapses of times that the buffer of Balloon A remains at the same usage, due to the break of the federation. Moreover, Balloon A does never reach the maximum capacity of the service that it has offered (i.e., 50 packets). For that reason, it remains publishing the service.

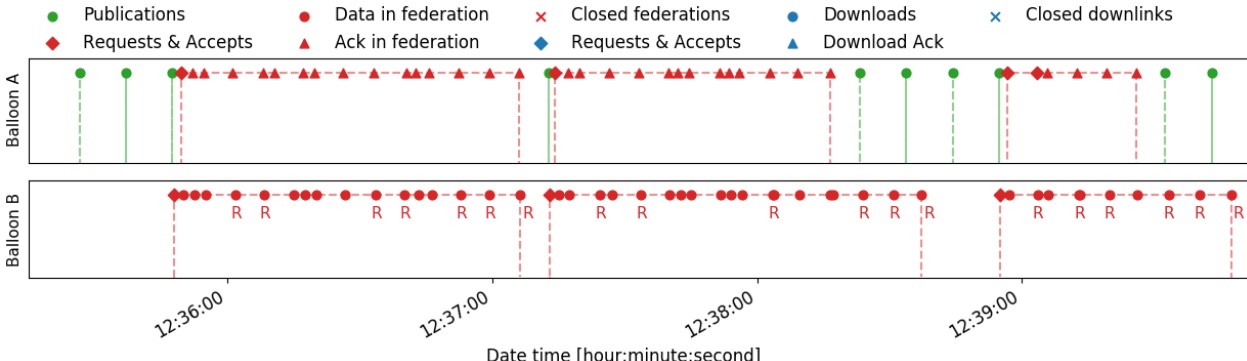

**Figure 19.** Sequence of packets exchanged in federations for storage.

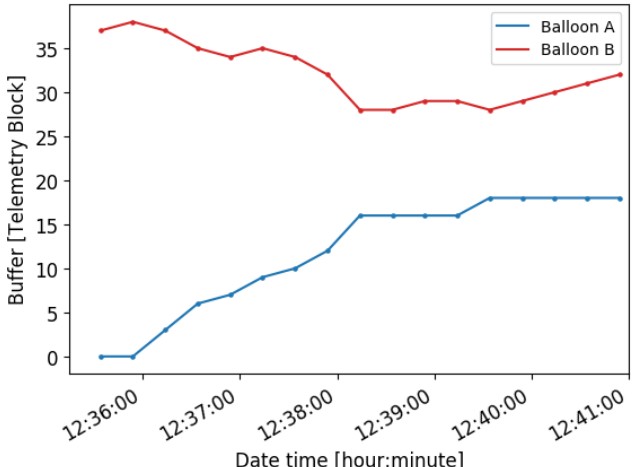

**Figure 20.** Buffer usage of Balloon A (blue) and Balloon B (red) during the federation for storage (data packet unit).

During this scenario, 19 data packets are exchanged between the balloons using federations. Taking into consideration the duration of the different federations, the achieved throughput is 107.6 bps. The PER in this scenario is smaller than in the other cases, with 42.2% of transmitted packets lost. Taking this link quality, the resulting throughput is equivalent to 94.4% of the maximum throughput (Equation (6)). Please note that the current communications must be compared with the reference values as a downlink communication because in this case there is no intermediate node that relays data.

Although the difficulties achieved to establish federations due to the poor quality of the link, multiple federations to share storage capacity have been achieved. These federations have enabled Balloon B to store its data into Balloon A. This exchange reduced the usage of Balloon B buffer, while the buffer in Balloon A was increasing. These results demonstrate that the proposed protocols can manage different types of services following the same interaction.

## 6. Conclusions and Future Research

Presently, the environmental, and socio-economic demands are requesting the development of new satellite applications that provides accurate, and near-real-time information. For that reason, new satellite architectures have emerged to satisfy these demands. The FSS paradigm promotes the collaboration between satellites, known as a federation, to share unused resources, such as storage capacity or downlink opportunities which are presently becoming critical in Earth Observation missions. However, satellite dynamics make that direct point-to-point communications (i.e., the ISL) are available during a specific lapse of time. This behavior limits thus the establishment of a federation to these periods [23].

The IoSat paradigm aims at addressing this situation by promoting the establishment of temporal networks between satellites following the opportunistic nature of the federations.

Both approaches pose multiple technology challenges that need to be solved before they can be used in the space. One of those challenges is related to the establishment and properly management of a federation. Please note that due to the opportunistic nature of the federation, traditional techniques cannot be applied directly, as in [16]. For that reason, this work has presented the protocol FeDeCoP, which enables the deployment of federations by executing an initial negotiation between the satellites. This negotiation becomes crucial because the amount of resources to be shared is agreed between the satellites. After this negotiation, the federation is established, and the resources consumed according to its type. The OSADP and the FeDeCoP protocols compose the federation protocol suite, which enables the deployment of federations in a network.

The proposed protocols have been evaluated in a realistic scenario with three stratospheric balloons. These balloons were launched sequentially to assess the capabilities of the protocols when balloons were on the ground, raising, and flying. The variability of the scenario helped to achieve the following conclusions: (1) the coverage of a ground station is enhanced employing the proposed protocols with federations as data relays; (2) the proposed protocols can balance the storage usage by sharing data among the balloons; and (3) the proposed protocols establish federations between ground devices and balloons to forward data, extending the original FSS concept to a hybrid network of terrestrial and non-terrestrial nodes.

The metrics retrieved from the campaign enable the quantification of the performance of the proposed protocols in a real environment. Among them, the PER metric characterizes the number of packets that the protocols cannot correct, and they are discarded or lost. It is driven by the quality of the communications channel. Despite the unexpected radio frequency interference which increased the PER, the proposed solution could establish federations thanks to its ARQ procedure. In all cases, the FeDeCoP protocol was able to detect and correct erroneous bits that allowed the correct delivery of more than a third of the data packets. Its robustness against channel effects has been demonstrated. In contrast, the RTT of the packets increases due to the large number of retransmissions. The resulting absolute throughput remained always bounded between 96.2 bps and 184.7 bps for download-based federations, and 107.6 bps for storage-based ones. Despite these absolute values may not be suitable for current satellite needs, if they are evaluated considering the channel environment (reflected on the PER) the achieved throughput was enclosed between 49.2% and 63.0% of its maximum value for download-based federations, and 94.4% for storage-based ones. This concludes that the proposed solution can achieve at least half of its maximum throughput depending on the channel conditions. As a summary, the FeDeCoP provides a reliable communication while trying to achieve the maximum throughput.

Despite these characteristics, this campaign also helped to identify **future enhancements** of the protocol. Regarding the throughput, other ARQ techniques can be applied to optimize the available capacity of the channel, such as Go-back-N or selective ARQ. Although the absolute values of the achieved throughput (hundred of bps) may not be suitable for satellite applications, the percentage of measured throughput with respect to its maximum value encourage to apply FeDeCoP with devices that provide larger data rates. In this case, the absolute performance could have potential interest for satellite data. Additionally, the results present large RTT values in federations due to retransmissions and the existence of an intermediate node. This indicates that the implementation of the protocol may not be optimal. Specifically, the use of no-real-time threads has been key on this behavior, which must be improved in future implementations.

Finally, the use of an omnidirectional antenna in the balloon limited the maximum communication range due to its gain, and the reception of interferent signals. This type of antenna was selected to mitigate the rotation of the balloon. In this case, the separation of a control and a data planes could benefit. Combining decoupled and independent planes it is then possible to isolate network managing tasks and traffic, which is used to optimize the data transmission on the other plane. Therefore, the control plane could provide omnidirectional patterns with long range techniques and low data rates. Meanwhile, the data plane would ensure large data rate and long range using directive antennas. The benefits of using this configuration for satellite federations has already been highlighted in a previous work [15].

Different conclusions and future approaches for FeDeCoP have been extracted from the campaign. Table 2 summarizes them by highlighting the relevant concepts of this manuscript. This stratospheric balloon campaign has become a demonstrator of the FeDeCoP and OSADP capabilities to deploy federations. These results encouraged to integrate the FSSExp payload with the OSADP and FeDeCoP protocols in the FSSCat mission [24]. This mission aims at demonstrating the benefits of applying the FSS concept to EO missions. In this mission, multiple payloads are distributed in two 6U CubeSats, known as $^3$Cat-5/A and $^3$Cat-5/B include the Flexible Microwave Payload-2 [25] a dual L-band microwave radiometer and GNSS-Reflectometer, and HyperScout-2 [26] a VNIR and TIR compact hyperspectral imager. which orbit following a train formation (i.e., one after the other). Additionally, to those EO payloads, both spacecraft include as technology demonstrator an Optical ISL (OISL) for CubeSat platforms [27] and the FSSExp payload. On 3 September 2020 both satellites were correctly launched, and they are in the operational phase. Apart of this CubeSat mission, the FeDeCoP and OSADP protocols need still to be studied in other scenarios in which considerable transmission time may impact their performance, such as interactions between Low Earth Orbit (LEO) and Geostatonary (GEO) satellites. The achieved results will be followed by future research to deploy federations in space.

**Table 2.** Summary of the conclusions.

| Results Achieved | Future Research |
|---|---|
| Success criteria achieved: federations for downlink contacts and storage capacity. | Study different ARQ mechanisms. |
| The FeDeCoP protocol worked in a challenging channel. | Use a high data rate device. |
| Appropriate throughput according to the experienced PER. | Use two different communication planes. |
| Federations enabled communication with out-of-range balloon. | Implementation with real-time threads. |
| RTT impacted by implementation and interferent signals. | |
| Federation concept with terrestrial and non-terrestrial nodes (hybrid case). | |

**Author Contributions:** All authors have read and agreed to the published version of the manuscript.

**Funding:** This work has been (partially) funded by "CommSensLab" Excellence Research Unit Maria de Maeztu (MINECO grant MDM-2016-0600), the Spanish Ministerio MICINN and EU ERDF project "SPOT: Sensing with pioneering opportunistic techniques" (grant RTI2018-099008-B-C21/AEI/10.13039/501100011033), by the grant PID2019-106808RA-I00/AEI/FEDER/UE from the EDRF and the Spanish Government, and by the Secretaria d'Universitats i Recerca del Departament d'Empresa i Coneixement de la Generalitat de Catalunya (2017 SGR 376, and 2017 SGR 219).

**Informed Consent Statement:** Informed consent was obtained from all subjects involved in the study.

**Acknowledgments:** The authors would like to acknowledge in-kind financial support by Skolkovo Institute of Science and Technology (Skoltech), with the Systems Engineering class of the Skoltech Space Center, and the Universitat Politècnica de Catalunya (UPC).

**Conflicts of Interest:** The authors declare no conflict of interest. The funders had no role in the design of the study; in the collection, analyses, or interpretation of data; in the writing of the manuscript, or in the decision to publish the results.

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
