# Peer review of "Towards Federated Satellite Systems and Internet of Satellites: The Federation Deployment Control Protocol"

_remotesensing, doi:10.3390/rs13050982_

Round 1

Reviewer 1 Report

The paper is well written. Unfortunately, the topic is marginally related to remote sensing research or application. It seems more appropriate to submit it to a telecom journal.

Author Response

Dear reviewer,

I appreciate your efforts in reading and reviewing the manuscript. After carefully thinking about the manuscript submission, we have read the scope of the special issue "Satellites Communications" of MDPI Remote Sensing. The Special issue "aims at bringing together multiple facets of next-generation satellite communication networks, systems and applications". The fields that motivated us to submit the manuscript were "novel communication system architecture" and "intersatellite links". This is the reason why we submitted in this special issue of the journal.

I hope that this explanation may clarify our selection.

Best,

Joan A. Ruiz-de-Azua

Reviewer 2 Report

This article is interesting. Introduction is too long. The summary of the article requires improvement and shortening to the most important conclusions. Remote Sensing magazine is not appropriate, better send your work to e.g. Sensors.

Author Response

Dear reviewer,

I appreciate your efforts in reading and reviewing the manuscript. We have modified the Introduction and the Conclusions section in order to be more accurate and highlight the important concepts that we wanted to remark.

Furthermore, after carefully thinking about the manuscript submission, we have read the scope of the special issue "Satellites Communications" of MDPI Remote Sensing. The Special issue "aims at bringing together multiple facets of next-generation satellite communication networks, systems, and applications". The fields that motivated us to submit the manuscript were "novel communication system architecture" and "inter-satellite links". This is the reason why we submitted in this special issue of the journal.

I hope that this explanation may clarify our selection.

Best,

Joan A. Ruiz-de-Azua

Reviewer 3 Report

Manuscript ID: Remote Sensing (ISSN 2072-4292)

Title: Towards Federated Satellite Systems and Internet of Satellites: The Federation Deployment Control Protocol.

  1. Overall I find the work interesting. Abstract however needs to be more concrete. The ‘Abstract’ must be revised and improved, in order to be better correlated with the content of the paper and objectives.
  2. Please consider that the results should represent the most important part of the ‘Abstract’. You should address briefly the obtained results.
  3. Line 43 to 81 some important references are missing
  4. The references must be prepared using instructions for authors.
  5. Discussion is expected to be more intense, focused on comparisons with other approaches and authors.
  6. Conclusions need to improve
  7. The research design need to improve
  8. Your research article brings highly interesting and important data concerning Towards Federated Satellite Systems and Internet of Satellites: The Federation Deployment Control Protocol. I do appreciate the variety of analytical instruments for the Federation Deployment Control Protocol applied in your project. I would like to congratulate the authors for the effort and scope of the article. It presents an interesting topic and has high readability and interest to readers. Regarding the manuscript, the current form needs major revisions. Some aspects need to improve (ex. abstract, introduction, and conclusion).

Author Response

Thank you very much for the review of the manuscript, your comments proposed really interesting enhancements. In order to structure our reply, I try to address each of the review points one-by-one:

1 - Overall I find the work interesting. Abstract however needs to be more concrete. The ‘Abstract’ must be revised and improved, in order to be better correlated with the content of the paper and objectives.

The Abstract has been entirely modified. Efforts have been done to focus on the results and the manuscript, reducing the introduction and motivation part of the abstract.

2 - Please consider that the results should represent the most important part of the ‘Abstract’. You should address briefly the obtained results.

The Abstract has been entirely modified. Efforts have been done to focus on the results and the manuscript, reducing the introduction and motivation part of the abstract.

3 - Line 43 to 81 some important references are missing

We have added additional references that we considered essential.

4 - The references must be prepared using instructions for authors.

References have been reviewed and modified following the instructions for authors.

5 - Discussion is expected to be more intense, focused on comparisons with other approaches and authors.

The manuscript presents just the single case in which authors tried to propose a communications protocol stack to deploy federations. This case is presented and discussed in the Introduction section, highlighting the drawbacks of its design. In particular, it is based on current Internet and Delay Tolerant Network technologies, but it does not consider the need for resource negotiation. Is indeed this gap that motivated the development of the FeDeCoP protocol, and its proper evaluation in a stratospheric balloon campaign. Due to different circumstances, we could not include the proposed protocol stack (the one with DTN and Internet) in the field campaign.

6 - Conclusions need to improve

The entire section that concludes the manuscript has been modified. It has been structured to present at the beginning the results of the campaign and the main conclusions retrieved from it. Then, future enhancements of the protocols are suggested.

7 - The research design need to improve

We have tried to clarify the use of the proposed metrics in the corresponding section. Additionally, the introduction has been modified to highlight the motivation of the proposed protocol.

Round 2

Reviewer 1 Report

The authors have modified their manuscript to highlight the role and importance of a protocol to help Earth Observation missions being more effective in data collection and re diffusion.

My recommendation is to publish as is

Reviewer 3 Report

I would like to congratulate the authors for the effort and scope of the article. It presents an interesting topic and has high readability and interest to readers. Regarding the manuscript, the current form accept for publication.